# The Immunological Role of the Placenta in SARS-CoV-2 Infection—Viral Transmission, Immune Regulation, and Lactoferrin Activity

**DOI:** 10.3390/ijms22115799

**Published:** 2021-05-28

**Authors:** Iwona Bukowska-Ośko, Marta Popiel, Paweł Kowalczyk

**Affiliations:** 1Department of Immunopathology of Infectious and Parasitic Diseases, Medical University of Warsaw, 02-091Warsaw, Poland; ibukowska@wum.edu.pl; 2Department of Animal Nutrition, The Kielanowski Institute of Animal Physiology and Nutrition, Polish Academy of Sciences, Instytucka 3, 05-110 Jabłonna, Poland; mszabl@wp.pl

**Keywords:** COVID-19, lactoferrin, pregnant women, oxidative stress, mother’s placenta

## Abstract

A pandemic of acute respiratory infections, due to a new type of coronavirus, can cause Severe Acute Respiratory Syndrome 2 (SARS-CoV-2) and has created the need for a better understanding of the clinical, epidemiological, and pathological features of COVID-19, especially in high-risk groups, such as pregnant women. Viral infections in pregnant women may have a much more severe course, and result in an increase in the rate of complications, including spontaneous abortion, stillbirth, and premature birth—which may cause long-term consequences in the offspring. In this review, we focus on the mother-fetal-placenta interface and its role in the potential transmission of SARS-CoV-2, including expression of viral receptors and proteases, placental pathology, and the presence of the virus in neonatal tissues and fluids. This review summarizes the current knowledge on the anti-viral activity of lactoferrin during viral infection in pregnant women, analyzes its role in the pathogenicity of pandemic virus particles, and describes the potential evidence for placental blocking/limiting of the transmission of the virus.

## 1. Introduction SARS-CoV-2 Infection

Coronaviruses (CoV) are single-stranded RNA viruses belonging to the order Nidovirales, family Coronaviridae and subfamily Coronaviridae [1]. In November 2019, a new type of coronavirus was identified in the Chinese city of Wuhan. It was named SARS-CoV-2, due to respiratory infections (COVID-19) it caused [2]. Several genetically different types of SARS-CoV-2 have been distinguished so far [3]. Clinical manifestations of respiratory infections vary from asymptomatic, mild upper and lower respiratory tract infection to life-threatening pneumonia with acute respiratory distress syndrome (ARDS) [4,5].

### Receptor Recognition Is the First Step of Viral Infection That Determines a Cell/Tissue Tropism

SARS-CoV-2 S-protein recognizes angiotensin-converting enzyme 2 (ACE2) [6,7], and by attaching to it may bind to other proteins: dipeptidyl peptidase 4 (DPP4), glucose regulated protein 78 (GRP78), carcinoembryonic antigen-related cell adhesion molecule 1 (CEACAM1), aminopeptidase N (APN), and recognize various sialosides and different glycosaminoglycans (GAGs) as cellular targets via lectin-type interactions. The cell entry of SARS-CoV-2 is possible after protein S activation by cellular proteases, including transmembrane serine protease 2 (TMPRSS2), cathepsin L, and furin [7,8,9]. TMPRSS2 and ACE2 co-expression is observed in several tissues, such as nasal epithelial cells, lungs, and bronchial branches [10,11].

SARS-CoV-2 infection begins in the nasal epithelial cells that initiate genetically innate and adaptive immune responses [10]. After cell invasion, a virus is recognized by the host’s innate immune system through pattern recognition receptors (PRRs), including C-type lectin-like receptors; toll-like receptor (TLR), NOD-like receptor (NLR), and RIG-I-like receptor (RLR) [12,13,14]. PRRs recognize molecules frequently found in pathogens’ pathogen-associated molecular patterns (PAMPs), or molecules released by damaged cells damage-associated molecular patterns (DAMPs), [15]. In addition, SARS-CoV-2 infection can cause host cell pyroptosis and the release of DAMPs. TLRs activation by DAMPs further enhances inflammation [15]. Consequently, the production of several anti-viral substances is activated, such as: “Lactoferrin” (LF), type I and III interferons, nitric oxide, b-defensins, and other chemokines and cytokines that recruit inflammatory cells, i.e., dendritic cells (DC), macrophages and influence adaptive immunity [16]. During SARS-CoV-2 infection, both Th1 and Th2 immunity pathways are activated almost at the same time during the infection course [17].

The course of COVID-19 varies significantly through the patients, strongly depending on immune responses. Elevated IL-6 levels are correlated with an increased risk of death in COVID-19 patients [18]; whereas, early activation of adaptive immunity is predicted for less disease development [19]. Moreover, patients with more severe disease (“cytokine-storm”) have increased plasma concentrations of interleukin (IL)-2, IL-7, IL-10, tumor necrosis factor α (TNFα), interferon-γ-inducible protein 10 (IP-10), granulocyte-colony stimulating factor (G-CSF), chemoattractant protein 1 (MCP-1) and macrophage inflammatory protein 1 alpha (MIP-1 alpha) [17]. The number of T-cells, such as helper T-cells, and memory helper T-cells, are decreased; whereas, naïve helper T-cells are increased in severe cases of COVID-19 compared to the group with mild symptoms [20]. The clinical outcomes, immune responses, and immunopathology are summarized in Table 1.

## 2. COVID-19 during Pregnancy—The Role of the Placenta

Pregnancy is a unique immunological state reflected by a combination of signals and responses from the maternal and the fetal–placental immune system. The maternal immune responses actively change throughout gestation from an anti-inflammatory state (first and third trimester) to pro-inflammatory state (second trimester) [21]. The immunological events show precise timing, named “immune clock” [22], and ensure the maintenance of maternal tolerance to fetus and protection against infectious agents. Physiological changes make pregnant women more susceptible to viral droplet-transmitted infections [23]; however, the virus infection appears to be milder compared to those caused by SARS-CoV and MERS-CoV [24,25] or H1N1 influenza [26]. The most common symptom of COVID-19 in pregnant women is fever (68%), then persistent dry cough (34%), malaise (13%), dyspnea (12%), and diarrhea (6%) [27]. An important role as a natural physical and immunological barrier that protects the fetus from various pathogens plays placenta [28,29,30,31]. In response to maternal infection, hypoxia, or nutrition status, it releases anti-microbial peptides and cytokines that activate and modulate maternal and placental immune response [32,33,34]. The effects of COVID-19 on the fetus are still largely unknown. The neonates’ outcomes following maternal COVID 19 may be diverse and sequel from immune-mediated events or direct cytopathic effect of the virus [35] (Figure 1).

Pregnant women who were diagnosed with COVID-19 are 20 times more likely to die than healthy pregnant women. These are the conclusions of a global study published in the medical journal JAMA Pediatrics [36]. The study, led by doctors from the University of Washington’s School of Medicine and the University of Oxford, analyzed data from 2100 pregnant women from 43 maternity hospitals in 18 countries between April and August 2020. Additionally, the risk of a severe infection was greatest in women with obesity, high blood pressure, or diabetes. For each pregnant woman with COVID-19, the researchers selected two pregnant women who were cared for in the same hospital and at the same stage of pregnancy, but with no known viral infection for comparison. They then followed both groups—706 women with COVID-19 infection and 1424 women without infection—until delivery and after discharge from the hospital. Eleven women from the group with COVID-19 died, and only one died in the group without COVID-19. In contrast, pregnant women with asymptomatic or mild infections were not at increased risk of intensive care, preterm labor, or pre-eclampsia. In the study, about 40 percent of pregnant women had COVID-19. The study authors found that women with COVID-19 have between 60 to 97 percent higher risk of premature birth. In women with COVID-19 who have a fever and respiratory failure, they found a five-fold increase in neonatal complications, including lung immaturity, brain damage, and visual disturbances. Of the babies born to mothers with COVID-19, eleven percent tested positive for the coronavirus. However, infections passed on to babies do not appear to be related to breastfeeding. Rather, the examination links them with delivery by cesarean section. These results highlighted the importance of including pregnant women in priority groups for vaccination against SARS-CoV-2 [35,36].

### 2.1. Maternal Immune Changes during Pregnancy

The SARS-CoV-2 (COVID-19) pandemic is still in the early stages of research, and preliminary case reports of infections in pregnant women are available. Changes in the hormone levels during pregnancy can modulate immune responses against pathogens [37]. Innate immunity remains unchanged, while adaptive responses change during pregnancy and vary with gestational age.

Innate immunity provides interaction with fetal tissues to promote successful placentation and pregnancy course, as well as is the first line of host defense against infections [38]. The maternal immune phenotype is characterized by an increase in peripheral blood neutrophils (up to 60–95%) monocytes, DCs (producing interferon (IFN) l), and suppression of peripheral NK cells in number and function [39,40,41]. The neutrophils directly interact with other immune cells, such as macrophages, DCs, NK cells, B, and T cells, therefore up- or down-modulating both immunities—innate and adaptive [42].

In opposite, cytotoxic CD8^+^ adaptive immune responses are diminished, bypassed, or even abrogated; whereas, regulatory immunity is enhanced in pregnant women. Moreover, a Th2 (pro-inflammatory) to Th1 (anti-inflammatory) cytokine shift is observed. Promotion of Th1 humoral responses can result in an altered clearance of infected cells [43].

### 2.2. Inflammatory Response to SARS-CoV-2 during Pregnancy—Infection Outcomes

Immune characteristics among pregnant and non-pregnant women with COVID-19 seem to be similar [17,44]. During virus infections, an increased Th2–associated cytokines profile is observed [17,45]. It is feasible that elevated Th2 immunity during pregnancy seems to be associated with a milder virus infection course [44]. Similarly, Th1 pro-inflammatory pathway inhibition, probably decreases the “cytokine storm” and results in COVID-19 severity being similar in pregnancy and non-pregnancy [46]. Pregnant women compared with non-pregnant women showed milder or no symptoms [47]. Moreover, the TLRs alteration during virus infection enhances immune response, however, it is not known how pregnancy affects this aspect of the viral response [48]. Importantly, the immune system activation influences clinical outcomes of the virus in mothers, as well as modulates the scale of fetal complications [49]. However, the risk of adverse clinical outcomes in pregnant women with the virus is still unclear. At present, there are insufficient data on the possible impact of the virus in early pregnancy, and only a few reports are available showing conflicting information: Hydrops fetalis and fetal deaths in one case [50], and no significantly increased risk of pregnancy loss in the second case [51]. Most studies concern pregnant women in the third trimester, and the observations are divergent from a similar clinical course of the disease [52] to an increased mortality rate among pregnant women compared to the rest of the population [53,54,55]. The pregnancy raises the morbidity of the virus, especially in the presence of risk factors, such as advanced maternal age, obesity, being Black or Hispanic, elevated D-dimer, and IL-6, as well as medical comorbidities [53,54,55]. The prevalence of cesarean sections in pregnant women with SARS-CoV-2 varied between 69.4% and 84.7% in different studies, the most common mentioned maternal complication was preterm labor (33.3%) [56], and a maternal mortality rate (MMR) reached 1.6% in some studies, while the others reported none or single deaths [46,54,55,57,58,59,60]. The COVID-19 related MMR in the UK was 1% (5/427 pregnant women) and in France was 0.2% (1/617 pregnant women) [61,62]. A significant increase in MMR has been documented in pregnant women from Brazil (12.7% vs. 5% of the general population) [63]. That high mortality rate may be a result of the low quality of prenatal care [63].

The maternal infection severity, including hypoxia or “cytokine storm”, may exaggerate the maternal immune system and participate in placental and fetal complications like fetal growth restriction (FGR) (10%), miscarriage (2%), preterm labor (39%) [64] (Figure 1). In addition, maternal inflammation during pregnancy can affect fetal brain development, CNS dysfunction, and behavioral phenotypes that may be recognized later in the postnatal life [65]. The fever, one of the most common symptoms of the virus, could be associated with increased hyperactivity disorder/attention-deficit later in life [65]. An elevated level of IL-6 observed in virus infection may be responsible for autism, psychosis, and neurosensorial deficits development in the offspring [66]. Similarly, increased maternal IL-17a levels correlate with autism spectrum-like phenotype in offspring [66]; whereas, an increased level of TNFα in the maternal peripheral blood additionally may have a toxic effect on early embryo development [67].

## 3. COVID 19—Placenta

### 3.1. The Maternal–Fetal Physical and Immunological Barrier

The placenta is essential organ with various physiological, immune, and endocrine functions needed—to nourish and protect the fetus. It is composed of cells from two different individuals—mother and fetus [68]. The fetal part of the placenta forms from the chorionic sac—including the amnion, yolk sac, chorion, and allantois. The outer layer of the placenta is called the trophoblast and consists of two layers: The cytotrophoblast layer (inner) and the syncytiotrophoblast layer (outer). The maternal part comes from the endometrium and is called the decidua with maternal vessels [69]. Between these two regions is located the intervillous space filled with maternal blood [70]. The basic functional units of the placenta are fetus-derived chorionic villi (CV) with fetoplacental vessels. CV are formed and maintained by the fusion of: syncytiotrophoblast (STB), extravillous trophoblasts (EVTs), and cytotrophoblasts (CTBs) [68]. The placenta’s unique structure and function determine the protective properties against most pathogens [71]. Its role in infections is multi-directional and involves: (1) Physical blockade of viral entry; (2) active anti-viral function and in case of infection (3) immunomodulatory action (Figure 2). The most important elements of the placenta as a physical barrier are: (i) A dense network of branch microvilli and periodical regeneration of the most outer STB layer [72], the lack of intracellular gap junctions between STB cells [73]; (ii) dense actin cytoskeletal network, forming a brush border at the apical surface of the STB layer [74]; (iii) limited expression of TLR or internalization receptors as E-Cadherin at the STB layer [75]; (iv) little to no expression of caveolins at STB surface [76]; (v) the basement membrane beneath the villous cytotrophoblast [77]. The immunological role of the placenta in infections depends on may things, including its immunomodulatory property with trophoblast-immune crosstalk. It has been suggested as a crucial component of the innate immune response. Immune cells from the fetal and maternal compartments interact to provide an intricate balance between fetal tolerance (pregnancy maintenance) and anti-microbial defense in case of infection. Moreover, the breakage or breach of the decidual or syncytial barrier continuity initiates a strong innate immune reaction against pathogens. The maternal decidua is composed of stromal cells and leukocytes (40% of decidua) [78]. 50–70% of decidual leukocytes are NK cells, 20–30% are macrophages, 10–30% are T cells, including regulatory T cells (Treg), and approximately 2% are DC’s [79,80,81]. The proportion of immune cells vary throughout pregnancy, with an increase in the proportion of T cells at term [82]. During the first trimester of pregnancy, macrophages and NK cells accumulate around the trophoblastic cells [83,84]. The fetal part of the placenta—the chorionic villi contains, at the core part, fetal macrophages (Hofbauer cells), fetal endothelial cells, fibroblasts, and mesenchymal stem/stromal cells (MSCs) [85,86,87]. The trophoblast releases several immunomodulatory molecules, such as a secretory leukocyte protease inhibitor (SLPI), β-defensins, and expresses “maternal lactoferrin” [88]. During pregnancy, TLRs (TLR-3, TLR-7, TLR-8, and TLR-9) are expressed on the surface of trophoblast, decidua, Hofbauer cells, endothelial cells, and chorioamniotic membranes. Furthermore, a soluble form of TLR2 is also present in amniotic fluid [88]. The expression of TLRs by trophoblast varies through the gestation (first trimester: villous cytotrophoblasts (vCTBs) and EVTs; term: STB and EVTs) [89]. Its immunoregulatory function includes caspase activation, cytokine production, and inflammatory response induction, as well as the release of anti-microbial peptides and proteins into the amniotic fluid [90]. They also play an important role in bridging innate and adaptive systems [91]. Acquired viral infections may disturb the immune regulation at the border of the mother and the fetus, leading to fetal damage, even when the virus does not reach it directly [92]. The TLR-3 receptor in the first trimester of pregnancy may mediate a rapid anti-viral response [93,94], and induce the production of cytokines, type I and III IFN [95]. Similarly, TLR7 induces the synthesis of anti-viral cytokines and play a role in preventing intrauterine transmission of some viruses (e.g., HBV) [96]. Cytokines and interferons are important mediators in healthy pregnancies, due to their role in regulating cell function, proliferation, and gene expression. However, their deregulation may disrupt the developmental paths of the fetus and placenta [97]. Lactoferrin may also play a similar role to TLR and interferon receptors. Moreover, to ensure the maternal humoral protection of fetus and neonates, the maternal antibodies are actively transported to the fetus via the neonatal IgG receptor expressed on the STB surface [98].

### 3.2. The Role of the Placenta in COVID-19

The role of the placenta in SARS-CoV-2 infections remains poorly understood and requires further research, including vertical transmission mechanisms, fetal infection, and its consequences. The most important activated molecular signaling pathways against viruses (including SARS-CoV-2) are: Type III IFN signaling, autophagy-regulating microRNAs, and the NF-κB pathway (summarized, in detail, by Kreis et al. [99]). At the placenta level, the type III IFN provides a powerful anti-viral response. The IFN initiates a signaling cascade that activates transcription of IFN-regulated genes. Given that SARS-CoV-2 induces the release of type III IFN, this could be one of the possible mechanisms protecting the fetus against SARS-CoV-2 infection. Trophoblastic micro RNAs may constitute an important placental anti-viral defense mechanism on viral invasion restriction and trophoblast integrity maintenance [99]. Therefore, using a miRNA construct as one of the therapeutic targets or as a vaccine against SARS-CoV-2 was proposed [100]. On the other hand, inhibition of the nuclear factor kappa-light-chain-enhancer of activated B (NF-κB) pathway in COVID-19 mice led to a reduction of inflammation and lung pathology in infected animals [101,102], showing the importance of NF-κB pathway regulation for a controlled immune response [103]. The immunomodulatory action of placental immune cells may cause immune response mitigation, cytokine storm reduction, damages of cells, tissue limitation, and reduction of SARS-CoV-2 transmission (Figure 2).

Information on the effects of similar viruses on the embryo and fetus is very limited. Very recently, humanity has faced two severe diseases caused by viruses from the same viral family as SARS-CoV-2. In both cases, the route of spread of the infection and clinical symptoms were very similar to COVID-19, but those diseases were associated with higher mortality. In February 2003, SARS (Severe Acute Respiratory Syndrome), caused by the SARS-CoV virus also began in China, and the virus has spread to nearly 30 countries. More than 8000 people fell ill then, 770 of them died. Among the cases reported in the literature, 17 cases concerned pregnant women, of which 12 were the largest group, while the remaining reports referred to single cases. The second disease was MERS (Middle East Respiratory Syndrome), caused by the MERS-CoV virus. The disease first appeared in Saudi Arabia in 2012, then in other countries of the Arabian Peninsula, including the USA, and in 2015 in South Korea. So far, about 2500 people have fallen ill with MERS, more than 860 have died. MERS has been reported in 13 women at different stages of pregnancy. Both in the case of SARS and MERS infection, spontaneous abortions, premature births, and the birth of healthy children were observed in pregnant women. Because the observed groups of women were sparse, the percentage data were not provided in the literature. The effects of SARS-CoV-2 on the embryo and fetus are investigable in those countries with congenital disability registers. Such defect register also operates in Poland under the name of the Polish Register of Congenital Developmental Defects (PRWWR), which was established in 1997 and covered the entire country, being the register subjected to the Polish Ministry of Health. PRWWR is the largest register of defects in the EU and has been in the EUROCAT register consortium since 2001. PRWWR is conducted jointly by doctors of many specialties, especially neonatologists, clinical geneticists, and pediatricians. One of the important goals of defect registers is the constant monitoring of possible mutagenic and teratogenic threats in the population for the purpose of quick identification and elimination of detected harmful agents. Keeping the Registry for over 20 years, it makes it possible to identify well the frequency and structure of congenital malformations in the Polish population. In the case of SARS-CoV-2 infection in pregnant women, the question of whether it also poses a threat to the developing child could be answered. If a pregnant woman becomes ill with COVID-19, it is important to avoid prolonged high body temperature, especially during the first trimester of pregnancy. It should be noted that increased stress in the mother adversely affects the developing child and even the future children of that child. This is mediated through epigenetic mechanisms. The first 12 weeks of pregnancy are a special period—when all the organs of the baby start to develop. Interfering factors, including teratogenic factors, can cause congenital disabilities and sometimes also disorders that are detected later in life. Teratogens include physical, chemical disruptors, certain drugs, biological agents, including some viruses, especially the rubella virus. Viral diseases during pregnancy can directly affect the embryo and the fetus, but also through the harmful effects of the increased body temperature of the sick mother. It should also be remembered that under normal conditions, approx. 12–25 percent of diagnosed pregnancies end in spontaneous miscarriage (normal population risk). In cases of miscarriage, the common cause is a chromosomal aberration in the embryo/fetus, which is a severe genetic disease of the developing baby, and is not associated with any infection. Similarly, 2–3 percent of children are born with a developmental defect (population risk). Thus, not every spontaneous miscarriage or neonate with defects born by a woman with SARS-CoV-2 should be associated with the effects of the virus [104,105,106,107,108].

### 3.3. The Possible Mechanisms of SARS-CoV-2 Vertical Transmission

Based on the current knowledge on viral vertical transmission routes, some possible mechanism used by the SARS-CoV-2 virus to cross the placenta [109] are proposed:(1)direct infection of STB syncytiotrophoblasts and their rupture, virion transcytosis via immune receptors ACE2 and Fc (FcR),(2)passage through endothelial microcirculation into the intravascular extravascular trophoblasts (EVT) or other placental cells, as well as(3)passing through infected maternal immune cells and(4)ascending vaginal infection (placental barrier) (Figure 2).

Placental tissue appears to be a potential site for SARS-CoV-2 infection, since the expression of the receptor and priming proteases in various cell types of the maternal–fetal interface was detected (1).

The presence of ACE2 was demonstrated in the female genital tract and the placenta, including STBs, vCTBs, invasive and intravascular trophoblast, vascular smooth muscle cells in primary villi, decidual cells, and vascular endothelial cells in umbilical vessels [99,110,111]. The expression of ACE2 dominates, especially in the early gestation placenta [112]. However, co-expression of ACE2 and TMPRSS2, by the human placenta and chorioamniotic membranes throughout pregnancy is rare [113]. The presence of alternative receptors and proteases for SARS-CoV-2 entry into STB cells has been suggested [114]. Recently proposed alternative receptors are DPP4 (CD26) and CD147 [115,116]. Whereas, furin and trypsin, both expressed on placental tissues through gestation, have been suggested as SARS-CoV-2 entry proteases [113,117,118].

Moreover, several placental cell types can be used as replication and entry sites of pathogens (2, 3): EVTs, vCTBs, Hofbauer cells, giant trophoblast cells, or maternal immune cells of decidua [87]. It is possible that PBMCs can be infected by SARS-CoV-2 and transmit the virus through the placenta, however, the viral replication does not seem to occur within this compartment [119].

### 3.4. Placenta Pathology

Maternal–fetal interplay during COVID-19 includes histomorphological changes in the infected placenta although, some research revealed SARS-CoV-2 presence in the placenta without abnormalities in placental histopathology [120]. Currently, several reports suggesting placental infection with SARS-CoV-2 and the viral presence were confirmed by PCR (placental tissue/amniotic membrane), immunohistochemistry, and in-situ hybridization assays (formalin-fixed paraffin-embedded tissue sections) [121,122,123,124,125,126]. The available findings of placental pathology from COVID-19 patients came from the third trimester [15,17,31,127,128], and the most common findings are vascular malperfusion (FVM), fetal vascular thrombosis and maternal vascular malperfusion (MVM) (20–73%), massive infection with generalized inflammation (presence of M2 macrophages, cytotoxic and helper T cells, and activated B-lymphocytes) (13–20%), fibrin deposition and intervillous thrombosis [15,17,31,127,128]. These abnormalities result from direct infection of cells, systemic inflammation (“cytokine storm”), hypercoagulable state, and maternal hypoxia [129]. Consequently, adverse perinatal outcomes: MVW associated intrauterine growth restriction (IUGR), increased incidences of preterm births, higher rates of perinatal death, miscarriage, pre-eclampsia, cesarean section deliveries are observed [130]. The placenta abnormalities seem to be independent of maternal clinical manifestation, and even asymptomatic pregnant women with viral infection may develop obstetrical complications [131]. Placental transmission of proinflammatory cytokines is likely to stimulate hormone signaling dysregulation, enhancing poor neonatal outcomes, due to oxygen deprivation [105,132].

### 3.5. The Vertical Transmission Rate

The virus present in the placenta does not determine the incidence of vertical transmission.

In most studies [133], detection of SARS-CoV-2 is performed using RT-PCR analysis on neonatal airway swabs, less common on placental tissue (30.0%), umbilical cord blood (32.5%), and amniotic cavity (reported in 35.0% of publications). The maternal diagnostic material includes additionally vaginal, cervical, or rectal swabs to detect genital tract viral shedding during vaginal delivery (22.5% of cases) [134,135,136,137,138,139,140,141]. In few studies, IgG and IgM serology in the mother and neonate was performed [135,137,142,143,144]. The neonatal SARS-CoV-2 infection was reported by Mahyuddin et al. in 25% of papers [145]; whereas, the rate of positive SARS-CoV-2 test for neonates born to mothers with COVID-19 was estimated by Jafari et al. as 8% [146]. Kotlayard et al. determined that viral transmission from mother to fetus may reach 3.2% based on the nasopharyngeal swab (NPS) RT-PCR testing. The rate of SARS-CoV-2 RNA positive test may occur in approximately 7.7% of placental and 2.9% of cord blood samples. The IgM serology confirmed SARS-CoV-2 infection in 3.7% of neonates [147]. The vertical transmission rate was estimated as 2.23–5.3% (1.3–16) [146]. Although the strong evidence that vertical transmission of the virus may occur; intrapartum transmission (exposure to maternal blood, vaginal secretions, or feces) and early postnatal transmission cannot be excluded. To date, Vivanti et al. [133] showed the clearest evidence for transplacental transmission of the virus, due to the detection of viral genetic material and protein in the placenta, and viral RNA alone in the amniotic fluid and neonatal blood sampled at birth.

The vertical transmission of the virus in the third trimester is approximately 3.2% (22/936) by infant NPS testing, with severe acute respiratory syndrome coronavirus 2 RNA positivity in other test sites ranging from 0% (0/51) in amniotic fluid and (0/17) urine, 3.6% (1/28) in the cord blood, 7.7% (2/26) by placental sample analysis, 9.7% (3/31) by rectal or anal swab, and 3.7% (3/81) by serology [147].

The vertical transmission risk seems to be relatively low. However, the lack of a precise and universal definition of the term “vertical” transmission prevents comparison of described cases of neonatal viral infection. Standardized definitions, including diagnosis time of neonates, method, and analyzed biological material clarified the rate of “vertical” transmission and distinguishing it between intrapartum and postnatal transmission of the virus. This may have implications for future research describing clinical courses and long-lasting post-infection neonatal outcomes. Moreover, further research and observations of pregnant women and their children with the virus are needed to assess further long-lasting clinical implications which can appear in offspring. Furthermore, more assessment should be made regarding the rates of vertical transmission in the early trimester of pregnancy and the potential risk for consequent fetal morbidity and mortality [135,136,137,138,139,140,141,142,143,144,145,146,148,149,150,151]. To date, research is underway to check whether the SARS-CoV-2 virus can be transmitted from mother to fetus. Until now, only a few cases of COVID-19 infection through the placenta have been documented, however, these occurred in the second and third trimesters of pregnancy. There are no known reports of the first trimester of pregnancy and infection of fetal tissues with the virus to date. Damage to the placenta and organs of the fetus from early pregnancy miscarriage was analyzed, related to the multi-organ hyperinflammatory process identified in histology and immunohistochemistry as a result of maternal COVID-19 infection. Analyzes were performed by immunohistochemical qPCR, immunofluorescence, and electron microscopy. The SARS-CoV-2 nucleocapsid protein, viral RNA molecules in the placenta and fetal tissues were found, accompanied by RNA replication revealed by positive immunostaining against double-stranded RNA (dsRNA). In this study, the results indicate that congenital SARS-CoV-2 infection is possible in the first trimester of pregnancy and that fetal organs, such as the lungs and kidneys, are targeted by the coronavirus [152].

In addition, the processes leading to damage to the placenta include thrombosis or vasculopathy that have been found in the placenta of women with COVID-19 infection. This is further evidence of the mechanisms of macrophage action by initiating anti-viral responses associated with chronic granulomatous (ulcerative) enteritis, which has been identified as a common feature of the virus-exposed placenta, supporting these hypotheses [153,154].

## 4. Maternal Lactoferrin (LF) in COVID-19—Pregnancy

Lactoferrin (LF, formerly known as lactotransferrin) is an iron-binding glycoprotein and a member of the transferrin family, with a molecular weight of around 80 kDa. It consists of a single complex polypeptide chain in two symmetrical spherical halves, and each of them can bind one iron ion [108,155].

LF is secreted by the glandular epithelium. The highest levels of LF are found in human colostrum, milk, and most exocrine secretions that wash mucosal surfaces. It is present in saliva, tears, semen, vaginal secretions, bronchial and nasal secretions, bile, pancreas, synovial fluid, urine, cerebrospinal fluid, and gastrointestinal fluids. It is also present in significant amounts in the secondary granularity of neutrophils (15 µg in 10^6^ neutrophils).

LF has an important biological role: The iron absorption and immune system action modulation are reflected in anti-microbial, anti-viral, antioxidant, anti-cancer, and anti-inflammatory functions. Because of numerous proven pro-health properties, LF has been used as a dietary supplement in many countries for over 40 years. Due to the availability of the raw material, bovine LF is used, which has a similar structure and properties to human LF [156]. Bovine LF (BLF) has been recognized by the US FDA, which gave it a GRAS status and EFSA as safe to be used as a dietary supplement and functional food additive [157].

### 4.1. LF Role in Host Defense—Anti-Viral Activity

Anti-viral effects of LF are widely studied in vitro and in several human clinical trials, which shed light on possible mechanisms of action, therapeutic efficacy, and safety.

Previous studies suggest that LF has huge anti-viral properties against: HPV, HSV-1, HSV-2, CMV, HIV, HBV, HCV, RSV, PIV, Hantavirus, coronavirus, rotavirus, polio, adenovirus, and potentially SARS-CoV too [158]. An indirect conclusion can be made on the role of LF in virus infection based on the study of gene expression in patients with the virus. The analysis of leukocytes in the peripheral blood of 10 patients in various clinical conditions showed a significant increase (150-fold) in the expression of genes encoding factors involved in the inflammatory reaction and LF [159].

In one clinical trial, healthy women taking oral lactoferrin observed a much less severe viral infection causing colds and inflammation of the stomach and intestines. In other clinical trials, oral lactoferrin was effective in relieving symptoms of viral gastroenteritis, including those caused by rotaviruses and noroviruses. Oral lactoferrin reduced the incidence and severity of symptoms of these diseases. Viral gastroenteritis was about four times more common in people taking 100 mg of lactoferrin once a week than those taking it six or seven times a week. In a 2011 study, lactoferrin was effective in blocking SARS coronavirus from invading host cells in in vitro model [160,161,162,163,164,165,166,167].

Due to the high genetic similarity (75%) of SARS-CoV and SARS-CoV-2, it can be assumed that LF will also inhibit the infection with the virus. A clinical trial with the use of BLF in people with SARS-CoV-2 showed a beneficial effect in most subjects and the reduction in the intensity of such symptoms of infection like: Headache (in all subjects), cough (in 50% of people vs. 61.11% baseline), myalgia (44.44% vs. 66.67%), fatigue/weakness (66.67% vs. 94.44%) [168].

LF role in host defense against viral infection results from inhibition of viral entry into target host cells, inhibition of viral replication, and immune response regulation after infection [160,162,163,164,165,166,167,169,170,171].

#### 4.1.1. LF as Virus “Entry Blocker”

Depending on the virus type, LF prevents infection of the target cell by (1) interfering with the attachment factor, or (2) binding to host cells into which the virus enters using them as receptors or co-receptors (i.e., glycosaminoglycan heparan sulfate (HSPG), ACE2, sialic acids, etc.), or (3) by binding directly to viral particles.

Until now, it is revealed that LF interacts with HIV-1 receptors CXCR4 and CCR5, and influenza A haemagglutinin (HA), blocking the viral entry process [169,170]. Moreover, it weakens the binding of Dengue virus (DENV)-2 to the host cell membrane by interacting with HSPG, a dendritic 3 cell-specific intercellular adhesion molecule, non-integrin capture (DC-SIGN), and low density lipoprotein receptors (LDLR) [163]. LF also can reduce hepatitis B virus (HBV) infection and replication, as well as hepatitis C virus (HCV) replication [171,172].

One of the LF possible anti-viral activity against SARS-CoV-2 is the inhibition of viral binding to the target cell surface in the early phase of virus amplification in the salivary glands, pharynx, and upper respiratory tract [162] (Figure 3).

At the N-terminus of the LF, there is an alkaline region that interacts with cell glycans (glycosaminoglycans, sialosides), used by many CoVs either as receptor determinants or as attachment factors, for docking to the mucus layer [162,173,174]. Two important classes of binding glycans are sialosides (SIA), which contain sialic acid (SA), and glycosaminoglycans, like heparan sulfate (HS). LF may inhibit/block attachment and accumulation of SARS-CoV-2 on the host cell membrane by occupying HSPGs and/or SIAPGs [162,175]. It also may bind to ACE2 receptor and blocks the initial interaction between virus and host cells [162]. Moreover, the interaction between LF and the three proteins present on the SARS-CoV-2 membrane, i.e., the spike (S), membrane (M), and envelope (E) proteins, is possible too [175].

**Figure 3 ijms-22-05799-f003:**
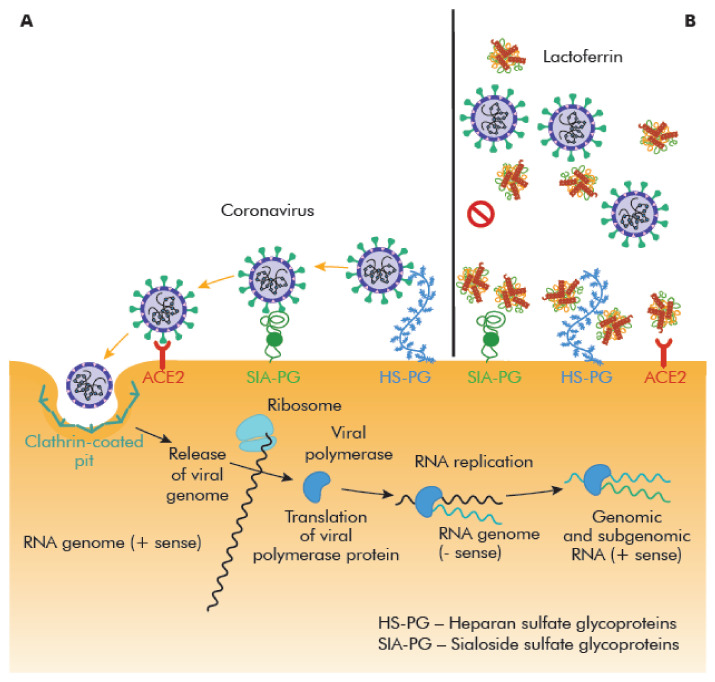
Lactoferrin effects on SARS-CoV-2 docking to cell surface receptors (CSR): Sialoside glycan (SIA) and heparan sulfate (HS). (**A**) SARS-CoV-2 attachment and entry to host cell. SIA and HS chains present on membrane glycoproteins (PGs) facilitate the attachment of the virion to the cell surface. Then the resulting complex binding to the cellular receptor (ACE2), initiating the process of internalization. (**B**) Human lactoferrin was found to exhibit anti-viral activity against SARS-CoV-2 infection, likely mechanisms of action are: Interaction with one of the proteins in the virion envelope S, M, or E, or competition for binding to glycan chains and/or probable the ACE2 receptor [162,175].

#### 4.1.2. LF Role in Anti-Viral Immune Response Modulation

LF plays an important role in the host’s first line of defense in numerous virus-induced infections. As a part of the innate immune defense, it regulates the activity of numerous immune cells of innate and adaptive response: Monocytes, macrophages, DCs, neutrophils, mast cells, NK cells, B, and T cells (Table 2) [158,176,177]. It has a broad immunoregulatory and anti-inflammatory effect, which may indicate its potential in the anti-viral treatment and prevention of SARS-CoV-2 infection [108,158].

LF modulates several APC pathways, including cell migration, activation, and antigen presentation, influences the expression of soluble immune mediators, affects the regulation of anti-inflammatory and immune responses (Figure 4) [158,176,177,178]. During inflammation, LF secretion increases dramatically, due to neutrophil degranulation [179]. Literature data indicate that LF increases the phagocytic activity of macrophages via binding to the specific receptors present on macrophages’ surface [180,181]. It increases the level of IL-12 in macrophagocytes, which attracts macrophages to inflammatory sites and activates CD4^+^ T cells [176]. On the other side, LF increases NK cell activity and stimulates neutrophils aggregation and adhesion [182].

LF also induces a response in which both types of T and B cells participate, thus increasing the response of specific antibodies against various specific antigens [183]. Oral administration of LF can increase the secretion of IgA and IgG in the intestines [184], reduce the number of leukocytes infiltrating the bronchoalveolar lavage fluid during viral infection with H1N1 influenza. LF increases the transcription of important immune-related genes, and their activation promotes the host’s systemic immunity [185]. These modulating effects on APC suggest a potential role for exogenous LF in enhancing adaptive immunity against COVID-19 infections. Th1 and Th2 cells lead to increased activity of macrophages in the intracellular elimination of pathogenic microorganisms [186], which leads to a reduction in the activity of T lymphocytes. This reduces the release of IL-5 and IL-17 cytokines, which prevents an excessive inflammatory response [164].

Moreover, LF increases the expression of type I IFNs (IFN-α/β), anti-viral cytokines, and immunomodulators that lead to the production of bioactive compounds and cytokines inhibiting viral replication [178,187].

### 4.2. LF Role during Pregnancy

The multi-directional action and safety of use mean that LF can be successfully used by pregnant women as part of the prophylaxis and/or treatment of many ailments related to pregnancy: LF has been known as a particle with proven efficacy and safety in the prevention and treatment of anemia. It can interact with both maternal and fetal micro-environments, creating physical and immunological barriers to avoid pathogenic microbes. LF supports the immune system in multiple ways: By inhibiting the growth or killing of pathogen cells, by regulating the activity of the immune system, and by manifesting prebiotic effects.

The greatest amounts of LF can be found in the milk of a nursing mother. The highest concentration of this valuable protein can be found in colostrum, i.e., the first food after childbirth, in which may reach the amount of approx. 6–8 g/L. However, mature milk contains smaller amounts of it, about 2–4 g/L. Considering the high concentration of LF in human milk, it can be concluded that it is extremely important for the proper development and protection of the newborn. It can significantly reduce the risk of infection because of pathogens when the baby’s body is not yet sufficiently protected by its immune system [188].

The results of some studies confirmed the importance of LF as a regulative factor of the immune response. It has been shown that even a small dose of LF (10–20 mg) stimulates the immune system [189]. Too strong inflammation, resulting from the imbalance of pro-inflammatory and anti-inflammatory cytokines, is a serious threat to pregnancy and may result in fetal growth restriction and premature delivery. Therefore, it can be assumed the balance restoration between pro- and anti-inflammatory stages will be protective. LF has such activity. This protein is physiologically present in the reproductive organs, also during pregnancy. It protects the mother and the fetus against infection and inflammation. The anti-inflammatory effect of LF is based on the reduction of the level of IL-6 in the blood plasma and cervicovaginal secretions. A high level of IL-6 causes shortening of the cervix and premature rupture of fetal membranes by stimulating the synthesis of prostaglandin F2α. A few studies showed that also, due to its anti-inflammatory effect, LF has a beneficial effect on reducing the risk of preterm labor [190].

The results of studies indicate that LF is a regulator of systemic iron metabolism in pregnant women [191]. Maternal lactoferrin activates signaling pathways that scavenge free radicals, regulate oxidative stress and pro-inflammatory cytokines in the Fenton reaction [192]. Iron sequestration by LF reduces oxidative stress [193]. Therefore, the role of LF in redox reactions during pregnancy and postpartum in women with the virus is justified. A study with anemic pregnant women using a preparation with iron-enriched with LF showed a greatest increase in ferritin levels and hemoglobin in comparison to the group using preparation only containing a high dose of iron. The same study also showed that using an iron supplement with LF increased the duration of pregnancy by an average of 1.5 weeks compared to the group taking iron alone [194,195]. In women taking BLF, pregnancy lasted longer [196]. Moreover, LF normalized the composition of the vaginal microbiota, cervical tension, extinguished local inflammation, regulated the level of pro-and anti-inflammatory factors (including cytokines, metalloproteinases, and prostaglandins), and protected against oxidative stress, which translated into the overall improvement of the clinical condition of patients and prolonged pregnancy that resulted in delivery on time [197,198,199]. Maternal LF, as a regulator of redox homeostasis, may play a pivotal role in the clinical management of COVID-19 during pregnancy.

LF forms a barrier between mother and fetus as a multi-functional regulator of the immune response with a broad spectrum of activity [197,198,199]. Regulation of the effects of LF on inflammatory mediators plays a key role in developing clinical therapies for the consequent cytokine storm and severe infection effects on pregnancy during COVID-19.

Detectable levels of LF appear in the amniotic fluid as early as 20 weeks of pregnancy. LF levels are elevated around week 30 and remain high until delivery. Neutrophil granularity also contributes to its elevation [200]. LF activates human growth hormone (hGH), and compared with epidermal growth factor (EGF), the action of LF is more pronounced in its effect on epithelial cells of the small intestine and endometrial proliferation [201]. LF levels increase during SARS-CoV-2 infection. STB also stimulates the release of LF and amniotic factors [202] by interacting with the mother’s and fetal microenvironment in the amniotic fluid and cervical mucus, which protects pregnant women against viral infections. Inflammatory cytokines, in particular IL-6, increase during amniotic membranes (CAM) infection, while LF levels are effective in inhibiting it [203]. High LF levels during pregnancy play an important protective role in reducing arterial hypertension by regulating access to the ACE2 membrane receptor, while inhibiting SARS-CoV-2 entry into cells [108].

### 4.3. Course of COVID-19 in Pregnant Women

Data on the course of COVID-19 in pregnant Polish women are very limited. During the SARS and MERS epidemics, pregnant women were more exposed to severe infections. However, no such relationship was found for the virus. Each of the newborns born to sick patients tested negative for the presence of coronavirus (data as of 17 March 2020). It is also unclear whether a pregnant mother would endanger the health and life of the child when she was pregnant. The Polish Society of Gynecologists and Obstetricians recommends that patients, before visiting the maternity hospital, undergo an epidemiological examination to determine the risk of exposure to the virus, which is to prevent the spread of infection to other pregnant women and medical staff [204]. Therefore, it is recommended that family deliveries in hospitals be suspended until further notice. Additionally, in every hospital with a maternity ward, there should be a separate so-called epidemiological emergency room, modeled on rapid prenatal diagnosis departments. This will enable basic examinations of the patient to be performed in compliance with the epidemiological regime. It is recommended that obstetricians limit the number of pregnant women’s visits to a minimum, preferably only to acute cases, and increase the possibility of contact by phone or e-mail. If a pregnant woman shows symptoms of infection, she should go to an epidemiological center to rule out virus infection in the first place. If the suspicion is high, the patient is treated as potentially infectious and should be referred to the so-called dedicated maternity hospital. The same should be done in an emergency to minimize the risk of virus transmission. In the absence of detailed studies on the course of viral infection in pregnant women, the management method does not have the level of EBM. The physician in each case should determine the benefit-risk ratio of the medical procedure, which is significantly increased during the pandemic [205].

The SARS-CoV-2 (COVID-19) pandemic is still in the early stages of research, and preliminary case series of infections in pregnant women are available. Recent press reports and scientific publications describe the negative effects of the virus on the placenta of women in the third trimester of pregnancy, describing the various degrees of fibrin deposition, which resulted in need of ending a pregnancy by emergency cesarean section. Three cases were described in which fibrin deposits occurred both inside and around the villi with local growth of syncytial nodules in the first case, multiple villous infarctions in the placenta (in the second case), and angioma (in the third case). Nucleic acid samples for the virus were collected from all three placentas and found to be negative [120]. In another study, where 16 placentae from women with the virus were analyzed, an increase in the frequency of MVM features, especially decimal arteriopathy, including severe development of atherosclerosis, extensive fibrinous necrosis, and excessive hyperplasia of the membranous artery, was observed [206]. Disorders of maternal hypertension, including gestational hypertension and over-developed pre-eclampsia, are the major risk factors for developing MVM [207], although only one patient was diagnosed with hypertension in this study. It is important to note that neither acute inflammatory pathology nor chronic inflammatory disease increase patients with the virus were compared to the control group [206].

At present, there are insufficient data on miscarriages in women with the virus, except in one case of a pregnant woman who had a second-trimester miscarriage. A fetal autopsy was negative for the virus and bacterial infection. SARS-CoV-2 was also absent in the fetal organs, such as lungs, liver, and thymus. Only inflammation, consisting of neutrophils and monocytes in the subcutaneous space, and excessive fibrin deposition in the intercellular space, was observed [208].

During the global SARS-CoV-1 epidemic, a significant increase in mortality and morbidity in pregnant women has been documented [209]. It has been found that the risk of viral pneumonia is significantly higher among pregnant women compared to the rest of the population [210].

Placental cells, trophoblasts, express TLRs, and their expression level varies depending on the age of the pregnant woman, trimester, and the stage of cell differentiation. Acquired viral infections may disturb the immune regulation at the border of the mother and the fetus tissues, leading to fetal damage, even when the virus does not reach it directly [92]. The TLR-3 receptor in the first trimester of pregnancy may mediate a rapid anti-viral response [93,94], and induce the production of cytokines, type I, and III interferon [93]. Also, TLR7 expressed in trophoblasts may induce the synthesis of anti-viral cytokines and preventing against HBV [95]. Cytokines and interferon also are important mediators in healthy pregnancies, due to their role in cell function regulation. However, their deregulation may disrupt the developmental paths of the fetus and placenta [97]. Lactoferrin may also play a similar role to TLR and interferon receptors.

At present, vaccines from many well-known concerns are already available, and guidelines for the treatment and control of the disease are being developed. Lung infections induced by this virus in pregnant women may increase the risk of maternal and fetal mortality [211], leading to numerous complications, such as premature delivery and a low gestational age [212]. Blood tests in pregnant women have revealed regular markers of the virus, such as lymphopenia, neutrophilia, and elevated levels of C-reactive protein in pregnant women [211,212,213]. Some reports also confirmed increases in ALT, AST, and D-dimers [214,215,216]. An important report confirmed that three mothers developed anemia and dyspnea, a low platelet count that could potentially be a risk factor during cesarean delivery [214,215,216].

## 5. Conclusions

Pharmacotherapy during pregnancy is often unavoidable. The number of pregnant women requiring medication are steadily increasing, partly because of advanced diagnostics, partly because of the rising rate of cases of SARS-CoV-2 infections in this group. Understanding the functions of drug transporters in the placenta in the context of pathological conditions and civilization diseases accompanying pregnancy that require long-term treatment of the mother and/or fetus, and the role of lactoferrin itself as a substance with potential anti-viral activity will enable a more accurate characterization of the penetration and design of drugs (also other xenobiotics) through the placental barrier, thus enabling safer pharmacotherapy not requiring the use of glucocorticosteroids to avoid the transmission of SARS-CoV-2 from mother to fetus, which is believed to be the main cause of maternal vascular insufficiency. The occurrence of lactoferrin in two forms: holo-Lf and apo-Lf—which provides the maximum potential for binding Fe^3+^ ions, leading to the activation of macrophages by binding to surface receptors and modulating the activity of T lymphocytes in cells (CD4^+^), which significantly strengthens the immune system. Lactoferrin reduces the formation of inflammation in the mother and fetus by modulating the production of cytokines and ROS, which in turn reduces iron overload. Lactoferrin also inhibits the binding of proteoglycans, such as heparan sulfate, preventing the virus from effectively penetrating the body. Lactoferrin is a naturally occurring iron chelator can bind to several receptors used by coronaviruses, thus blocking their entry into the host cells. In this way, it may have immunomodulatory and anti-inflammatory effects, which means that it has a very high therapeutic value during the current COVID-19 pandemic.

Taking all the above into consideration, more research is still needed in the case of lactoferrin utility and influence on the infection course in pregnant women.

COVID-19 causes 6.9 million deaths worldwide, more than double the number reported in official reports, according to a new study by the Institute for Health Metrics and Evaluation (IHME). According to the institute’s calculations, nearly 150,000 have already died in Poland, due to COVID-19. The Institute for Health Metrics and Evaluation (IHME) is an independent institute for global health research at the Washington University School of Medicine. According to IHME, the adopted estimates are based on the methodology for measuring the global burden of disease, which the institute has been using for many years. The IHME estimated the total number of COVID-19-related deaths by comparing the projected number of deaths from all causes based on pre-pandemic trends with the actual number of deaths from all causes recorded during the pandemic. The IHME analysis found that in almost every country, the number of deaths from COVID-19 is significantly underestimated. The updated analysis shows that to date, more people in the United States have died from COVID-19 than in any other country, and that the total number of deaths exceeds 905,000. The institute points out that many deaths from COVID-19 are not reported, as country reports only include deaths occurring in the hospital or among patients with confirmed infection. In many places, this problem is exacerbated by ineffective health reporting systems and poor access to healthcare. When broken down by region, Latin America and the Caribbean, Central and Eastern Europe, and Central Asia were affected by the highest number of deaths. The modeling algorithm proposed by IHME is updated weekly and can be found at (covid19.healthdata.org (accessed on 21 May 2021)).

## Figures and Tables

**Figure 1 ijms-22-05799-f001:**
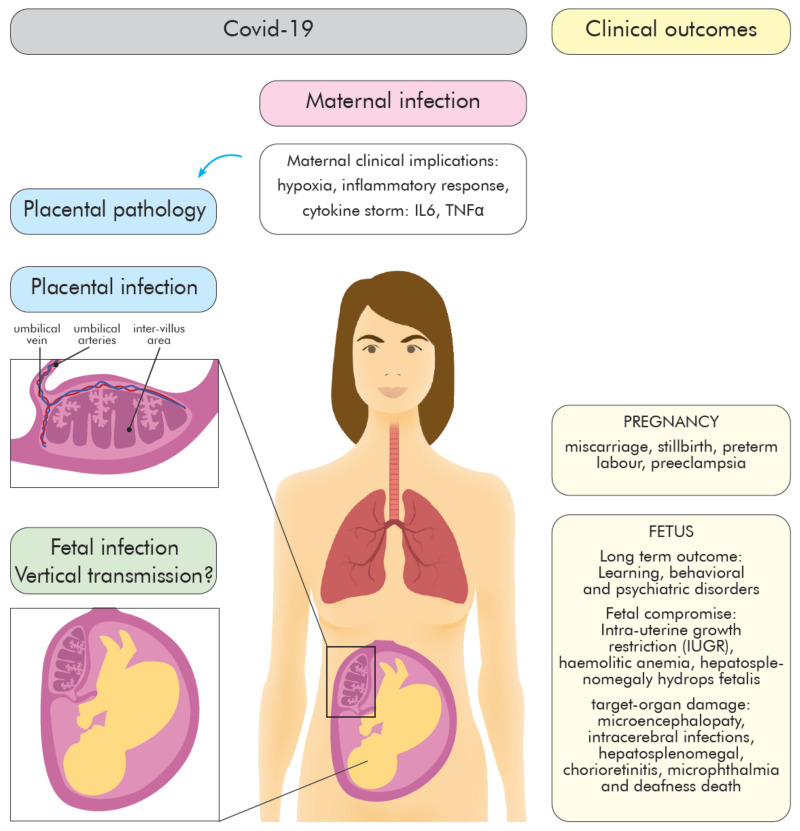
Implications of SARS-CoV-2 virus infection at the maternal–fetal interface. The placental function disruption (histomorphological alterations) as a consequence of maternal clinical implications of COVID-19 (e.g., hypoxia, cytokine storm) and/or placenta infection, as well as probable fetal infection (vertical transmission) may result in pregnancy complications, compromise fetal health and long term adverse neonatal outcomes.

**Figure 2 ijms-22-05799-f002:**
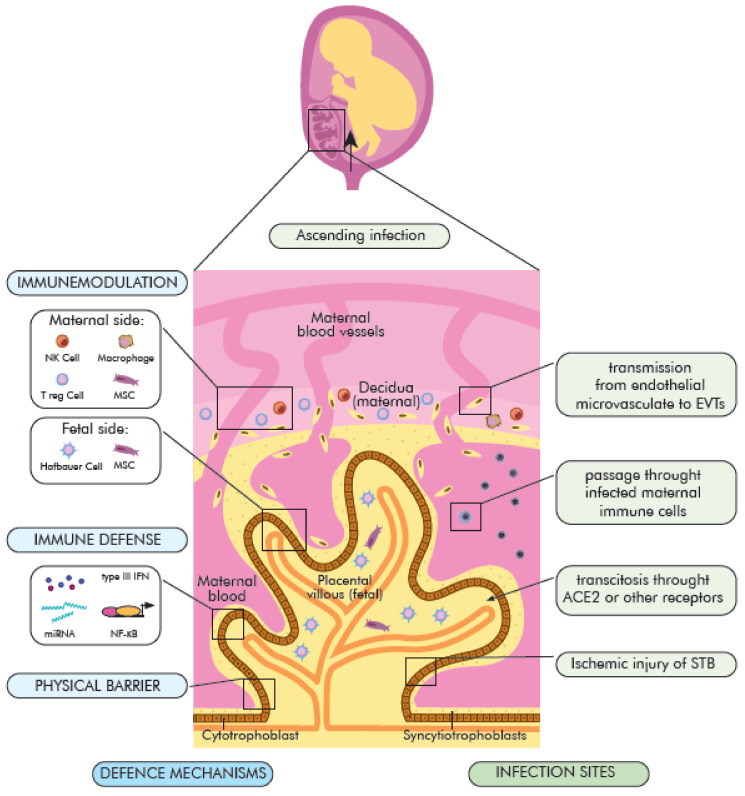
The defense mechanism of the placenta and potential infection sites of SARS-CoV-2. Placental properties that prevent SARS-CoV-2 infection include: Physical blockade, release/synthesis of anti-viral molecules (miRNA, IFN III, NF-κB), and stimulation of immune defense by decidual and fetal immune cells. The SARS-CoV-2 fetal infection may occur due to placental barrier breakage or via ascending route. Abbreviations: ACE2, Angiotensin converting enzyme 2; EVTs, extravillous trophoblasts; IFN, interferon; miRNA, microRNA; MSC, mesenchymal stromal cells; NF-κB, nuclear factor kappa-light-chain-enhancer of activated B cells; NK cell, natural killer cell; STB, syncytiotrophoblast; T reg cell, regulatory T cell.

**Figure 4 ijms-22-05799-f004:**
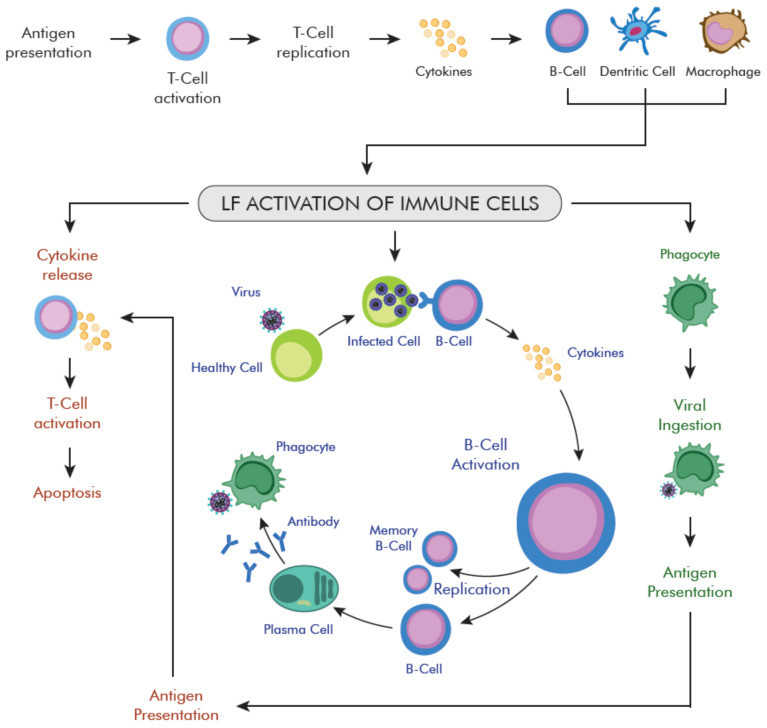
The lactofferin-mediated immune response of anti-viral cells [175]. Main activities and cites of action of lactoferrin (LF) in host defense against viral entry. The diagram illustrates the main effects of lactoferrin on T and B cells, macrophages, and dendritic cells (DC). In the initial stage of a specific immune response, antigen-presenting cells activate naïve T cells, presenting them with antigens and providing additional costimulatory signals. Activated T lymphocytes undergo intensive division and form functionally differentiated subpopulations: Effector T lymphocytes and memory T lymphocytes. Effector T lymphocytes migrate to the site of infection, where they inactivate the pathogens present there, and then die by apoptosis. B-cells capture foreign antigens and presents them to T-cells. Activated B-cells (after antigen binding) transform into antibody secreted plasma cells or memory B-cells. Lactoferrin modulates antigen-presenting cell action, including migration and activation, inactivates cells infected with the virus, activates B cells that destroy infected cells with the participation of T cells. Moreover, LF affects the expression of soluble immune mediators (cytokines, chemokines, and other effector molecules) to regulate inflammatory and immune responses.

**Table 1 ijms-22-05799-t001:** Immune characteristics related to the clinical course of SARS-CoV-2 infection.

Clinical Course of Infection	Asymptomatic/Mild COVID-19 (Appropriate Immune Response)	Severe COVID-19 (Defective Immune Response)
Immune response	Immune cell activation: Monocytes, neutrophils, B cells, CD4^+^ T, CD8^+^ T cells, and NK cells; Physical activation of Cytokines/Chemokines secretion: TNFα, IL-2, IL-6, IL-7, IL-8, IL-17, G-CSF; Virus inactivation by neutralizing antibodies;	Lowering of immune cells number: Monocytes, eosinophils, basophils, B cells, CD4^+^ T, CD8^+^ T cells, and NK cells; Increased number of neutrophils; Elevated levels of IL-6, IL2R, IL-10, and TNF-α; Increased level of SARS-CoV-2 specific IgG;
Immunopathology	None/Mild	Lymphopenia—infection susceptibility; Systemic cytokine storm; Lymphocyte dysfunction: T cell depletion and exhaustion; virus-specific T-cells central memory phenotype; Antibody-dependent enhancement (ADE) of infection;
Clinical outcomes	Infection resolution	ARDS Respiratory failure Multi-organ dysfunction Sepsis

NK cell, natural killer cell; IL, interleukin; TNFα, tumor necrosis factor α; G-CSF, granulocyte-colony stimulating factor; ARDS, respiratory distress syndrome.

**Table 2 ijms-22-05799-t002:** Pro-inflammatory and anti-inflammatory action of LF.

Pro-Inflammatory Action of Lactoferrine (LF)	Anti-Inflammatory Action of Lactoferrine (LF)
increasing B cell maturation and activation; activation of NK cells; promoting maturation and activation of antigen-presenting cells (APC) in T cell, i.e., monocytes, macrophages, DCs, and B cell; activation of phagocytosis (granulocytes, monocytes, macrophages, DC); stimulation of myelopoiesis; regulation of the balance between Th1 and Th2 immune response; induction of pro-inflammatory cytokines synthesis (e.g., IL-1, IL-2, IL-4, IL-6, IL-8, TNF-α, IFN-α, IFN-γ); induction of expression of costimulatory and adhesion molecules (e.g., ICAM-1, CD3, CD4, LFA-1, MHC class II) on the cell surface; induction of ROS production (e.g., H_2_O_2_, NO) that are toxic to microorganisms and regulate inflammation; promotion of immune cells adhesion to the vascular endothelium and chemotaxis to the sites of infection; activation of the complement system;	increasing of anti-inflammatory cytokines secretion (IL-4, IL-10, IL-11); accelerating the processes of repairing damaged tissues and wound healing (e.g., by activating angiogenesis). inhibition of pro-inflammatory cytokines production (IL-1, IL-6, IL-8, TNF-α); inhibition of B cell activity; inhibition of inflammatory mediators production: Prostaglandin E2 (PGE2) and cyclooxygenase-2 (COX-2); inhibition of adhesion and costimulatory molecules expression (ICAM-1, CD86, E-selectin) on the immune cells surface; inhibition of metalloproteinases production; inhibition of ROS formation; inhibition of mast cells and DCs activity;

## Data Availability

On request of those interested.

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
