# Peer review of "The Immunological Role of the Placenta in SARS-CoV-2 Infection—Viral Transmission, Immune Regulation, and Lactoferrin Activity"

_ijms, 2021, doi:10.3390/ijms22115799_

Round 1

Reviewer 1 Report

Very interesting paper. I appreciate a lot this paper. Figures make the paper high quality. The paper is difficult to improve because is very well wrote and high quality paper.

Only some minor suggestions and congratulation to the authors

  1. Introduction: update data on global burden of SARS CoV2
  2. The role of placenta: very well wrote. Please add better the fetal outcome
  3. Conclusion: give some publich health action that came from your paper with focus on low setting countries

Author Response

Reviewer 1

Thank you very much for all valuable comments that contributed to the improvement of the quality of our manuscript

The Reviewer pointed out that there are more details in the “Introduction”, “The role of placenta” and “Conclusions” sections needed. All sections have been extended as requested.

  1. Introduction: update data on global burden of SARS CoV2

The update on global burden of SARS-CoV-2 has been included in the description of the manuscript (page 4 – 5, line 110-132) and its description is provided below:

Pregnant women who were diagnosed with COVID-19 are 20 times more likely to die than pregnant women who are healthy. These are the conclusions of a global study published in the medical journal JAMA Pediatrics [36]. The study, led by doctors from the University of Washington's School of Medicine and the University of Oxford, analyzed data from 2,100 pregnant women from 43 maternity hospitals in 18 countries between April and August 2020. Additionally, the risk of a severe infection was greatest in women with obesity, high blood pressure or diabetes. For each infected pregnant woman, the researchers selected for comparison two pregnant women who were cared for in the same hospital and at the same stage of pregnancy, but with no known viral infection. They then followed both groups - 706 women with COVID-19 infection and 1,424 women without infection - until delivery and after discharge from hospital. Eleven women from the group infected with COVID-19 died, and only one died in the uninfected group. In contrast, pregnant women with asymptomatic or mild infection were not at increased risk of intensive care, preterm labor, or pre-eclampsia. In the study, about 40 percent of pregnant women were infected. The study authors found that infected women have between 60 to 97 percent higher risk of premature birth. In infected women with fever and respiratory failure, they found a five-fold increase in neonatal complications including lung immaturity, brain damage, and visual disturbances. Of the babies born to infected mothers, eleven percent tested positive for the coronavirus. However, infections passed on to babies do not appear to be related to breastfeeding. Rather, the examination links them with delivery by caesarean section. These results highlighted the importance of including pregnant women in priority groups for vaccination against SARS-CoV-2 [35, 36].

  1. The role of placenta: please add better the fetal outcome

The role of fetal outcome have been included in the description of the manuscript (page 19, line 730-750) and its description is provided below:

Information on the effects of similar viruses on the embryo and fetus is very limited. Very recently, humanity has faced two severe diseases caused by viruses from the same viral family as SARS-CoV-2. In both cases, the route of spread of the infection and clinical symptoms were very similar to COVID-19, but those diseases were associated with higher mortality. In February 2003, SARS (Severe Acute Respiratory Syndrome) caused by the SARS-CoV virus also began in China, and the virus has spread to nearly 30 countries. More than 8,000 people fell ill then, 770 of them died. Among the cases reported in the literature, 17 cases concerned pregnant women, of which 12 were the largest group, while the remaining reports refered to single cases. The second disease was MERS (Middle East Respiratory Syndrome) caused by the MERS-CoV virus. The disease first appeared in Saudi Arabia in 2012, then in other countries of the Arabian Peninsula, including the USA, and in 2015 in the South Korea. So far, about 2,500 people have fallen ill with MERS, more than 860 have died. MERS has been reported in 13 women at different stages of pregnancy. Both in case of SARS and MERS infection, spontaneous abortions, premature births, and the birth of healthy children were observed in pregnant women. Due to the fact that the observed groups of women were sparse, the percentage data were not provided in the literature. The effects of SARS-CoV-2 on the embryo and fetus are investigable in those countries with birth defect registers. Such defect register also operates in Poland under the name of the Polish Register of Congenital Developmental Defects (PRWWR), which was established in 1997 and covers the entire country, being the register subjected to the Polish Ministry of Health. PRWWR is the largest register of defects in the EU and has been in the EUROCAT register consortium since 2001. PRWWR is conducted jointly by doctors of many specialties, especially neonatologists, clinical geneticists and pediatricians. One of the important goals of defect registers is the constant monitoring of possible mutagenic and teratogenic threats in the population for the purpose of quick identification and elimination of detected harmful agent. Keeping the Registry for over 20 years, it makes possible to identify well the frequency and structure of congenital malformations in the Polish population. In the case of SARS-CoV-2 infection in pregnant women the question whether it also poses a threat to the developing child could be answered. If a pregnant woman becomes ill with COVID-19, it is important to avoid prolonged high body temperature, especially during the first trimester of pregnancy. It should be noted that increased stress in the mother adversely affects the developing child and even the future children of that child. This is mediated through epigenetic mechanisms. The first 12 weeks of pregnancy are special period - when all the organs of the baby start to develop. Interfering factors, including teratogenic factors, can cause birth defects and sometimes also disorders that are detected laterin life. Teratogens include physical, chemical distruptors, certain drugs, biological agents, including some viruses, especially rubella virus. Viral diseases during pregnancy can affect the embryo and the fetus in a direct way, but also through the harmful effects of the increased body temperature of the sick mother. It should also be remembered that under normal conditions, approx. 12-25 percent of diagnosed pregnancies end in spontaneous miscarriage (normal population risk). In cases of miscarriage, the common cause is a chromosomal aberration in the embryo / fetus, which is a severe genetic disease of the developing baby, and is not associated with any infection. Similarly, 2-3 percent of children are born with a developmental defect (population risk). Thus, not every spontaneous miscarriage or neonate with defects born by a woman infected with SARS-CoV-2 should be associated with the effects of the virus [104-108].

  1. Conclusion: give some public health action that came from your paper with focus on low setting countries.

Additional information in Conslusion section has been included in the description of the manuscript (page 4 – 5, line 110-132) and its description is provided below:

COVID-19 causes 6.9 million deaths worldwide, more than double the number re-ported in official reports, according to a new study by the Institute for Health Metrics and Evaluation (IHME). According to the institute's calculations, nearly 150,000 have already died in Poland due to COVID-19. The Institute for Health Metrics and Evaluation (IHME) is an independent institute for global health research at the Washington University School of Medicine. According to IHME, the adopted estimates are based on the methodology for measuring the global burden of disease, which the institute has been using for many years. The IHME estimated the total number of COVID-19-related deaths by comparing the projected number of deaths from all causes based on pre-pandemic trends with the actual number of deaths from all causes recorded during the pandemic period. The IHME analysis found that in almost every country the number of deaths from COVID-19 is significantly underestimated. The updated analysis shows that to date, more people in the United States have died from COVID-19 than in any other country, and that the total number of deaths exceeds 905,000. The institute points out that many deaths from COVID-19 are not reported, as country reports only include deaths occurring in the hospital or among patients with confirmed infection. In many places, this problem is exacerbated by ineffective health reporting systems and poor access to healthcare. When broken down by region, Latin America and the Caribbean, Central and Eastern Europe and Central Asia were affected by the highest number of deaths. The modeling algorithm proposed by IHME is updated weekly and can be found at (covid19.healthdata.org.).

Reviewer 2 Report

In the present manuscript, Bukowska and collaborators review the possible mechanisms of infection of the SARS-CoV-2 virus in the placenta and its possibility to be transmitted vertically. The authors make a good compilation of evidence on this phenomenon and give an approach towards using molecules such as lactoferrin and its potential use as an antiviral. However, the issue remains controversial as clinical evidence remains scant, and in vitro studies have not been performed. It is necessary to qualify the ideas presented in the section on vertical transmission mechanisms (lines 223 to 246). since the lack of experimental evidence can be subject to misinterpretation by the reader. For example,  the authors needed to document more the damage to the placenta, such as thrombosis or vasculopathy that has been found in the placentas of COVID-positive women. Thus giving more evidence to these mechanisms and macrophage infiltration associated with chronic histiocytic intevillositis that has been identified as a common feature in placentas exposed to the virus, this evidence could give more support to the hypotheses mentioned (PMID: 33764543, 33764543). Finally, consider it necessary for the figure captions to be more descriptive since the figures are not detailed, which can also be confusing if they do not put that description.

Author Response

Reviewer 2

Thank you very much for all valuable comments that contributed to the improvement of the quality of our manuscript

  1. The Reviewer brings attention that the issue (vertical transmission) remains controversial as clinical evidence remains scant, and in vitro studies have not been performed. It is necessary to qualify the ideas presented in the section on vertical transmission mechanisms (lines 223 to 246). since the lack of experimental evidence can be subject to misinterpretation by the reader. For example, the authors needed to document more the damage to the placenta, such as thrombosis or vasculopathy that has been found in the placentas of COVID-positive women. Thus giving more evidence to these mechanisms and macrophage infiltration associated with chronic histiocytic intevillositis that has been identified as a common feature in placentas exposed to the virus, this evidence could give more support to the hypotheses mentioned (PMID: 33764543, 33764543).

This information has been provided and suggested citations included as requested (page 11 - 12, line 414 – 432). Description is provided below. Moreover the issue presented in the section on vertical transmission mechanisms has been corrected and underline the hypothetical role of mentioned vertical-transmission routes for SARS-CoV-2 (3.3. The possible mechanisms of SARS-CoV-2 vertical transmission  section, page 10, line 327-329).

To date, research is underway to check whether the SARS-CoV-2 virus can be transmitted from mother to fetus. Until now, only a few cases of COVID-19 infection through the placenta have been documented, however, these occurred in the second and third trimesters of pregnancy. There are no known reports of the first trimester of pregnancy and infection of fetal tissues with virus to date. Damage to the placenta and organs of the fetus from early pregnancy miscarriage was analyzed, related to the multi-organ hyperinflammatory process identified in histology and immunohistochemistry as a result of maternal COVID-19 infection. Analyzes were performed by immunohistochemical qPCR, immuno-fluorescence and electron microscopy. The SARS-CoV-2 nucleocapsid protein, viral RNA molecules in the placenta and fetal tissues were found, accompanied by RNA replication revealed by positive immunostaining against double-stranded RNA (dsRNA). In this study, the results indicate that congenital SARS-CoV-2 infection is possible in the first tri-mester of pregnancy and that fetal organs such as the lungs and kidneys are targeted by the coronavirus [152]. In addition, the processes leading to damage to the placenta include thrombosis or vasculopathy that have been found in the placenta of women with COVID-19 infection. This is further evidence of the mechanisms of macrophage action by initiating antiviral responses associated with chronic granulomatous (ulcerative) enteritis, which has been identified as a common feature of the virus-exposed placenta, supporting these hypotheses [153, 154].

  1. Finally, consider it necessary for the figure captions to be more descriptive since the figures are not detailed, which can also be confusing if they do not put that description.

As suggested by the reviewer the captions under the figures have been extended.

Reviewer 3 Report

The article “The immunological role of the placenta in SARS-CoV-2 infection – viral transmission, immune regulation, and lactoferrin activity” is a comprehensive review about the connection between pregnancy and Sars-Cov-2 infection. The investigated topic is relevant, interesting, and the authors collected a respectable amount of information about it. However, there are issues that need to be addressed before publication.

A major issue is that the article is hard to read and understand. Extensive proofreading by an English language professional is needed and do not forget to include the figures too.

Another major issue is that the figure does not add much value to the article. Figure 1 does not contain much information (also, death is really the only option for the mother?). Figure 3 would be more suitable for a table, but it is arguable if necessary at all. Figure 4 does not explain much, which could not be understood from the manuscript (HS and SIA abbreviations are reversed). Figure 5 looks like a good description of how the immune response works, but there is no specific information how LF is connected to this process.
I think that there are several other complex mechanisms mentioned in this review, which could benefit from graphical presentation instead of the current figures.

The introduction of abbreviations is not consistent.

Do not forget to include author names in the final version.

The use of subheadings is inconsistent (e.g., there is no 2.1.2, so 2.1.1 is not needed).

Author Response

Reviewer 3

Thank you very much for all valuable comments that contributed to the improvement of the quality of our manuscript

  1. The Reviewer pointed out that extensive proofreading of the manuscript by an English language professional is needed.

The language will be formally corrected by the editorial board in the Medical University of Warsaw and/or the journal's editors, as needed, after manuscript approval by the reviewers.

  1. The Reviewer brings attention to the low value of figures included to the article: Figure 1 does not contain much information. Figure 3 would be more suitable for a table, but it is arguable if necessary at all. Figure 4 does not explain much, which could not be understood from the manuscript (HS and SIA abbreviations are reversed). Figure 5 looks like a good description of how the immune response works, but there is no specific information how LF is connected to this process.

All pointed figures and captions have been duly corrected. We extended the captions under the figures to be more descriptive.

Figure 1 schematically depics maternal-fetal interplay following SARS-CoV-2 infection. We wanted to present the complex and multidimensional relationships described in the manuscript regarding the fetal outcome in the course of COVID 19 in pregnant women in a simplified and unambiguous approach to the reader. For this reason, Figure 1 is not overloaded with content and is an introduction to the issues described in detail in the paper. In our intention, Figure 1 is a graphic introduction to the issues described in detail in the content of the manuscript. At the same time, we agree with the reviewer that the figure may be removed if necessary in order to increase the substantive value of the work. Moreover, Figure 3 has been removed. The manuscript sections 4.1. “LF role in immune response modulation” and 4.3. “LF as a protection against SARS-CoV-2 virus infection” has been modified and extended in accordance with reviewer suggestions and better illustrate the issues presented in Figures 4 (now Figure 3) and 5 (Figure 4). Moreover Figure 4 (now Figure 3) has been changed and abbreviations corrected.

  1. The Reviewer pointed out that the author names were not included, as well as introduction of abbreviations and the use of subheadings were inconsistent

We apologize for all inconvenience, all missing information has been provided and inconsistencies have been corrected. All the irregularities noticed by the reviewer arose at the stage of editing the manuscript by the editors. The manuscript we sent to the editor contained both the authors' data and a list of abbreviations on which the manuscript was written. We made the decision to not extend the abbreviations in the text with the aim of increasing the transparency of the issues described.

Round 2

Reviewer 3 Report

I have no further comments.